# Plastic Optical Fiber Spectral Filter Based on In-Line Holes

Azael Mora-Nuñez [1], Héctor Santiago-Hernández [1,*], Beethoven Bravo-Medina [1], Anuar Beltran-Gonzalez [1], Jesús Flores-Payán [2], José Luis de la Cruz-González [1] and Olivier Pottiez [3]

1 Departamento de Ingeniería Electro-Fotónica, Universidad de Guadalajara (UDG), Blvd. Gral. Marcelino García Barragán 1421, Guadalajara 44430, Jalisco, Mexico; dejesus.mora@academicos.udg.mx (A.M.-N.); salvador.bravo@academicos.udg.mx (B.B.-M.); anuar.beltran@academicos.udg.mx (A.B.-G.); jose.dgonzalez@alumnos.udg.mx (J.L.d.l.C.-G.)

2 Departamento de Ciencias Biomédicas, Universidad de Guadalajara (UDG), Av. Nuevo Periférico 555, Tonalá 45425, Jalisco, Mexico; jesus.flores9295@alumnos.udg.mx

3 Centro de Investigaciones en Óptica (CIO), Loma del Bosque 115, Col. Lomas del Campestre, León 37150, Guanajuato, Mexico; pottiez@cio.mx

* Correspondence: hector.santiagoh@academicos.udg.mx

**Abstract:** We propose a spectral filter based on a plastic optical fiber with micro-holes as a low-cost, robust, and highly reproducible spectral filter. The spectral filter is explored for two configurations: a fiber extended in a straight line and a fiber optic loop mirror scheme configuration. The transmission traces indicate a spectral blue shift in peak transmission, at 587 nm, 567 nm, 556 nm, and 536 nm for zero, one, two, and three holes in the fiber, respectively. The filter exhibits a bandpass period of approximately 120 nm. Additionally, we conduct a comparison of the transmission with holes separated by distances of 1 cm and 500 μm. The results demonstrate that the distance between holes does not alter the spectral transmission of the filter. In the case of the fiber loop mirror configuration, we observe that the bandpass can be adjusted, suggesting the presence of multimode interference. Exploring variations in the refractive index within the holes by filling them with glucose solutions at various concentrations, we determine that the filtering band and spectral shape remain unaltered, ensuring the stable and robust operation of our spectral filter.

**Keywords:** plastic optical fiber; fiber optical filter; fiber optic loop mirror





## 1. Introduction

Polymer or plastic optical fibers (POFs) are large-diameter, flexible, low-cost, and durable multimode fibers composed of different types of plastic material, including poly-methyl methacrylate (PMMA), polystyrene, polycarbonate, and perfluorinated materials [1,2]. Although optical fibers are generally composed of silica, POFs present the additional advantages of ease of fabrication and operation, a low cost, high flexibility, softness, a light weight, higher fracture toughness, higher strain limits, and biocompatibility when compared to silica optical fibers [3,4]. Such properties make POFs a relevant technology for applications such as sensors [5–7]; local area networks [1,8]; and the fabrication of devices such as tapers [9], connectors [10], couplers [11,12], gratings [13,14], and even interferometric systems [5,15].

Fiber optical filters play a crucial role in communication systems [16], particularly in wavelength-division multiplexing (WDM). They are also widely employed in spectroscopy [17] and fiber optic sensing [18,19]. There are two main types of commercially available fiber-based filters, namely fiber Bragg grating (FBG) filters and long-period grating (LPG) filters, both inscribed in the core of a single-mode fiber (SMF). FBG filters function as band rejection filters [20], whereas LPG filters have the flexibility to be reconfigured into a bandpass filter [21]. This is achieved by using two LPGs in series, facilitating the coupling of resonant light from the fiber core into the cladding and subsequently back into the core [22]. Nevertheless, these devices are predominantly manufactured using

silica fibers. In recent years, substantial research efforts have been directed towards the development of fiber Bragg grating (FBG) filters, exploring alternative materials and techniques [23,24]. Furthermore, LPG filters based on POFs for multiple sensing applications have been reported [25,26].

Generally, methods for POFs' structural modification include tapering [27], side polishing [28], screw-shaped [29], spiral-shaped [30], and multi-notched [31]. Moreover, there are very few studies of holes in the fibers [32–34]. Specifically, techniques to create holes in POFs involve the use of a drill bit [32,33] and a mechanical die press print method [34]. In this work, we take advantage of the valuable technique proposed in [32] to develop a bandpass filter based on multiple in-line holes in a POF. Our findings encompass the examination of the spectral transmission of the POF filter with multiple holes spaced by both centimeters and micrometers, employing both straight-line and fiber loop mirror schemes, respectively. The results demonstrate that incorporating multiple holes and adopting the fiber loop mirror (FOLM) scheme enables us to fine-tune the bandpass filter. Furthermore, we establish that variations in the refractive index within the holes do not alter the profile of the spectral response. However, the response intensity is sensitive to index changes, and we observe that the loop mirror configuration exhibits heightened sensitivity to refractive index changes at lower concentrations.

Some theoretical studies have demonstrated that morphological disorder in waveguides can significantly impact the transmitted optical signal. For instance, the simulations in [35] examined the transmission spectra of photonic components with up to four types of morphological disorder as a material parameter. The results showed that various types of disorder, as well as combinations of two types of disorder, can be intentionally utilized to adapt the spectral transmission characteristics. Consequently, these findings may inform the design of small-footprint photonic components such as scatterers, diffusers, and waveguides. Additionally, in [36], the researchers demonstrated that the light transport in a disordered ensemble of resonant atoms placed in a waveguide is highly sensitive to the cross-sectional size of the waveguide. Specifically, the transmittance magnitude was found to possess a complex, non-monotonic dependence on the transverse size of the waveguide, highlighting the intricate nature of light propagation in disordered systems.

The production of optical components based on fiber optic technology involves costly and labor-intensive processes, while silicon-based fibers tend to become delicate after processing. Consequently, plastic optical fibers present a solution to develop cost-effective optical components that are practical for domestic use. Although their application is confined to short distances, they prove highly suitable for smaller-scale systems like automotives [37], textile sensors [38], medical devices [6], or home networking [39].

## 2. Experimental Setup

The experimental setup is depicted in Figure 1, showcasing both the straight-line hole (a) and the in-loop mirror hole scheme (b). Both configurations are introduced to showcase the capabilities of the proposed spectrum filter utilizing perforated plastic fibers. The configuration involves a 1.5-m-long commercial LF-750 plastic optical fiber with a numerical aperture of 0.5. According to the technical specifications, the POF core and cladding materials are polymethyl methacrylate and a fluorinated polymer, respectively. The refractive index (RI) of the core is 1.49, while that of the cladding is 1.41, forming a step-index profile. The cladding diameter is 750 ± 45 µm, and the core diameter is 735 ± 45 µm. Perforations were created in the middle of the 1.5 m POF, with 1, 2, and 3 holes drilled separately with 500 µm or 1 cm intervals.

**POF with in-line holes on straight-line configuration**

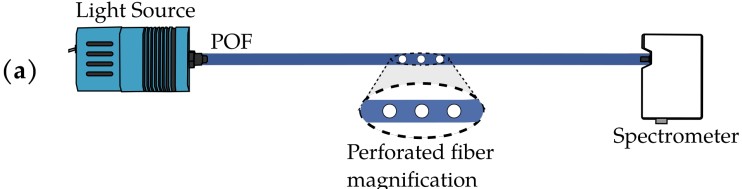

**POF with in-line holes on loop-mirror configuration**

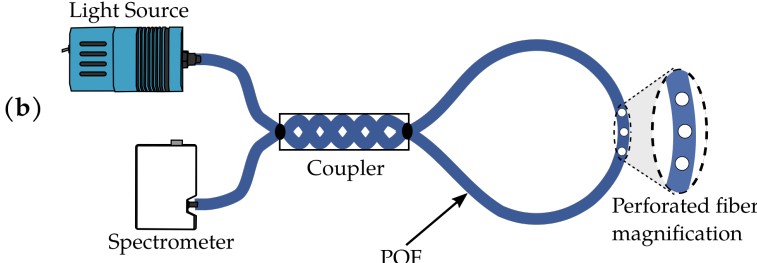

**Figure 1.** Schematic diagram of fiber optical filters: (**a**) straight-line configuration, (**b**) POFLM configuration.

The plastic optical fiber loop mirror (POFLM), shown in Figure 1b, comprises a drum with a 10 cm diameter, around which the 1.5 m POF is coiled. For the POFLM, the coupler manufacturing process follows the same procedure as described in [15]. However, in this study, additional environmental protection was provided to the coupler by enclosing it within a metallic tube. Subsequently, the transmission of several fibers and POFLMs with multiple perforations was measured, as depicted in Figure 1. The interrogation system consisted of a tungsten lamp that served as a broadband light source, as well as an spectrometer (Ocean Optics, model USB2000CXR, Dunedin, FL, USA) that aided in measuring the optical signals at the output of the filters. Thus, the characterization of the spectral filter was conducted by measuring the transmittance as a function of the number of holes.

Additionally, the sensing of the refractive index was examined by filling the holes with dextrose (D-Glucose, J. T. Baker, Avantor Performance Materials, S. A. de C.V., Xalostoc, Edo. de Mex. México) at concentrations ranging from distilled water to saturation. The specific dextrose solutions covered a density range of 0 g/100 mL (pure distilled water) to 33 g/100 mL, with intermediate concentrations at intervals of 3 g/100 mL. Moreover, introducing variations in the refractive index within the holes enabled us to discern the filter's response under diverse conditions. In essence, modifications in the frequency or intensity of the signal via variations in the refractive index allowed the enhanced utilization of the proposed spectral filter. The solutions, ranging from the lowest to the highest refractive index, were introduced into the holes. Each solution was carefully poured into a small container. Due to the challenging removal of the highest-index solution from the hole, extra care was taken to ensure accurate results. After measurement, the liquid was extracted using a syringe, absorbed with paper, and then dried by air blowing, ensuring that the output power matched that of the hole filled with air. This process was repeated for each solution.

The in-line sub-millimeter holes were created using a computer numerical control (CNC) mounted drill for 3D printing, as illustrated in Figure 2a. A 200-µm-diameter drill was employed in the process, with a commercial optical microscope (SHENZHENSHI SHENGY, Futian district, Shenzhen, China) used for the real-time monitoring of the perforation. Figure 2b–d illustrate the POFs with one, two, and three perforations, respectively.

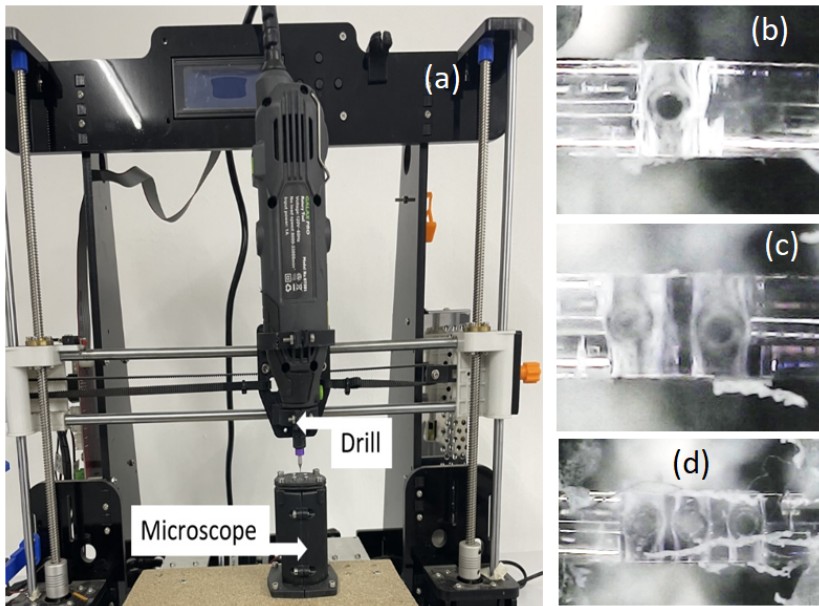

**Figure 2.** (**a**) CNC-POF perforation system: (**b**) one hole, (**c**) two holes, and (**d**) three holes. The holes are separated by 500 μm.

## 3. Results and Discussion

The transmission quality of an optical fiber is defined by two loss properties: attenuation and dispersion. The sources of attenuation in optical fibers can indeed be categorized into two types: intrinsic and extrinsic. Intrinsic sources are inherent to the material, such as material absorption and Rayleigh scattering, while extrinsic sources are introduced during the manufacturing of the fiber, including structural imperfections and microbends. Dispersion refers to the broadening of the signals caused by the wavelength-dependent speed of propagation within the optical fiber, and it imposes constraints on the transmission over extended distances.

A POF exhibits inherent losses attributed to factors such as absorption in the constituent material and Rayleigh scattering [40]. These losses arise from the molecular vibrational absorption of groups like C-H, N-H, and O-H; absorption due to electronic transitions within molecular bonds; and scattering resulting from fluctuations in composition, orientation, and density [41,42]. However, extrinsic losses can be introduced into POFs by altering their transmission through perforation, as proposed here for optical filtering.

The experimental results confirm that the spectral transmission of the POF is indeed modified by the presence of holes, as illustrated in the traces in Figure 3. These measurements, depicted in a straight line, as shown in Figure 1a, reveal two significant regions. The first region spans approximately 470 nm to 600 nm, where the spectral transmission experiences a blue shift with an increasing number of holes in the POF. In this region, at maximum transmission, the traces exhibit a blue shift of 587 nm, 567 nm, 556 nm, and 536 nm for zero, one, two, and three holes in the fiber, respectively. The second region, encompassing 600 nm to 710 nm, shows a decrease in transmission intensity as the number of holes in the POF increases. Specifically, at 650 nm (indicated by the dashed line in Figure 3), a second peak displays a decrease in transmission intensity at zero, one, two, and three holes and the respective transmission values (0.963, 0.792, 0.597, and 0.438). This peak also exhibits a modest blue shift as the number of holes is increased. Moreover, the first region indicates a bandpass of approximately 115.78 nm, measured as the full width at half maximum (FWHM). Considering both regions, the estimated period of the spectral filter is around 120 nm. It is essential to note that the traces are normalized, for the sake of comparison, as fibers with more holes exhibit a lower transmission intensity.

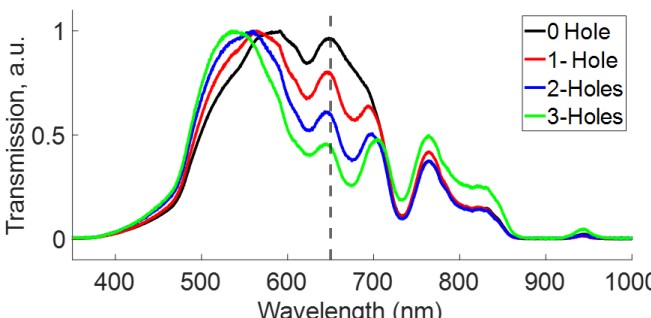

**Figure 3.** Transmission in straight line, with several holes separated by 1 cm. The vertical dashed line displays the 650 nm wavelength.

Total internal reflection (TIR) is the primary phenomenon governing the transmission of light in optical fibers. However, the presence of holes alters the conditions for TIR, as illustrated by Shin in [32], where the holes are considered as lenses with varying refractive indices in the POF. In our investigation, we measured the transmission through a 1.5 m section of a POF with several holes separated by approximately 500 μm, to assess the impact of the hole distance on our results. Figure 4 illustrates that the transmission of fibers with holes separated by micrometers closely resembles the transmission when the holes are spaced 1 cm apart, as depicted in Figure 3, showing a large blue shift in the left peak and a signal decay at around 650 nm. The results suggest that the elimination of the spectral components occurs solely due to the presence of holes. Furthermore, the number of holes allows for the tailoring of the transmission of the bandpass filter. Moreover, the region around 770 nm presents complex behavior that we attribute to the interaction of multiple modes [43,44].

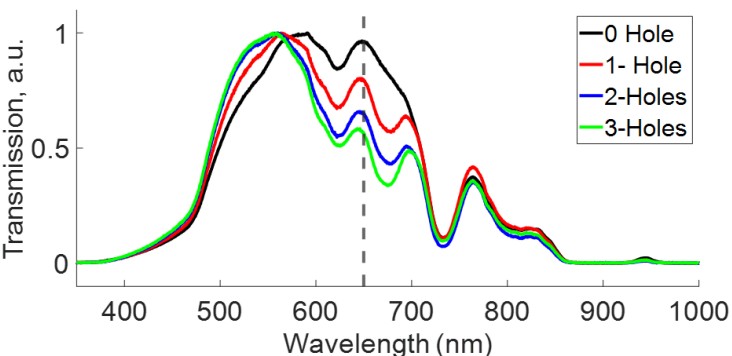

**Figure 4.** Transmission in straight line, with several holes separated by 500 μm. The vertical dashed line displays the 650 nm wavelength.

Moreover, leveraging the additional degrees of freedom in the mirror configuration compared to the straight-line setup, and taking advantage of the experience acquired in previous work [15], we analyzed the response with perforations in the loop of the POFLM, as shown in Figure 5. The first region (470–600 nm) similarly describes a blue shift in the traces when increasing the number of holes. However, the second region (600–710 nm) depicts variations in the transmission intensity. The transmission of the straight line (blue trace) depicts a minor intensity in the second region, in comparison with the POFLM (red trace) with one, two, and threes holes displayed in Figure 5a–c, respectively. As displayed in the same figures, the results suggest that the bandpass filter can perform elimination better in some spectral components in the straight-line scheme configuration than in the POFLM. In addition to the perforations, it is possible to note that the traces of the transmission for the POFLM configuration are very different, suggesting that the tuning of the bandpass is possible in the POFLM. Interestingly, for the POFLM with two holes, the maximum of the transmission shifts from the first to the second region. We attribute this filter tuning to the

multimode interference occurring at the output of the POFLM, producing a stable output due to the averaging effect over the modes, as reported in [45].

The transmittance of fiber loop reflectors based on single-mode fibers depends on the wavelength, coupling ratio, and loop birefringence [46–49]. However, in our setup, we use a multimode fiber that considerably modifies and complicates such dependence. Reference [45] theoretically shows that the presence of multiple modes and mode coupling in the fiber loop degrades the fringe visibility and produces a residual phase error that affects the sensitivity and bias stability. In our experimental study, we take advantage of the fact that variations in the refractive index in the loop of the Sagnac produce varying losses in the fiber core.

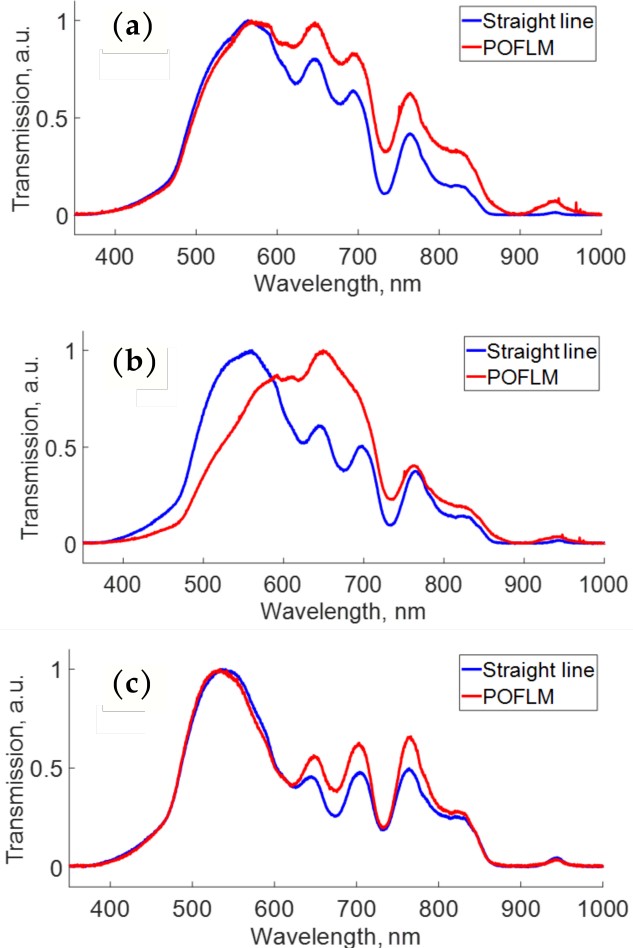

**Figure 5.** Transmission comparison between straight-line and POFLM schemes with (**a**) one hole, (**b**) two holes, and (**c**) three holes separated by 1 cm.

The fiber optic loop mirror is widely recognized as a simple and robust structure, created by forming a fiber loop between the output ports of a directional 2 × 2 coupler. To assess the impact of the curvature in the loop fiber within our setup, we conducted measurements on the transmission through a 1.5 m section of the fiber, both in a straight line and coiled on a drum.

Figure 6 illustrates that the spectral transmission remains unaffected by the curvature introduced by coiling the fiber on a drum with a 10 cm diameter. However, no significant variations in the transmission traces were observed within the range of 400 nm to 850 nm, with a minimal difference of 0.0414 in the intensity at 764 nm. From the observations presented in Figure 6, we can conclude that the curvature of the fiber in the drum does not alter the spectral transmission.

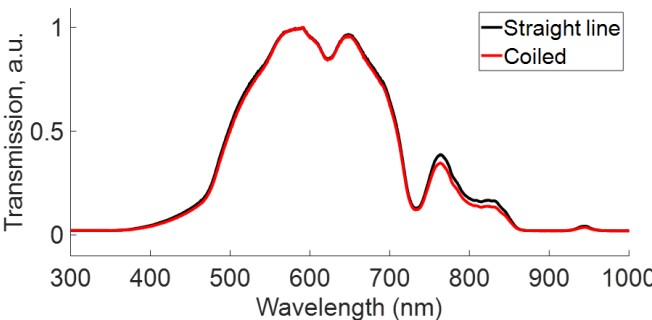

**Figure 6.** Comparison of transmission in a fiber extended in a straight line and a coiled fiber. Both fiber optic configurations without holes.

The optical directional coupler serves as a crucial component in the construction of a fiber loop mirror. For characterization purposes, we measured the spectrum at the output ports of a specially manufactured directional $2 \times 2$ coupler. The coupler was developed using the twisting technique of a POF along approximately 5 cm, as illustrated in [15]. It is essential to note that one port of the coupler serves as the input for light. The output ports are designated as follows: the output port of the active fiber (the same input fiber) is marked as T1, the output port of the passive fiber (which receives the signal from the active fiber) is marked as T2, and the output port where the signal is reflected by Fresnel reflection is marked as R. The spectra measured at each port are depicted in Figure 7.

In the same figure, notable differences are observed in the region of 470 nm to 600 nm. However, from 600 nm to 1000 nm, all spectra exhibit similar behavior. At 591 nm, the traces display an intensity of 1.0, 0.861, and 0.821 for output ports T1, R, and T2, respectively. The traces in the same figure indicate that the fiber optic filter is not affected by the coupler for wavelengths beyond 600 nm, as demonstrated in Figure 5. However, wavelengths below 600 nm exhibit differences in attenuation at each port, suggesting that the filter can be tuned in this region by the coupler.

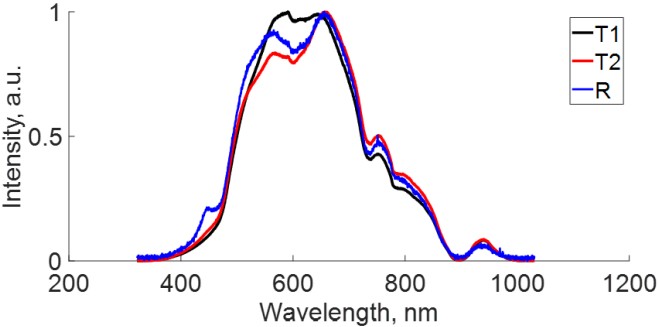

**Figure 7.** Influence of coupler in the spectral response at outputs T1 (active fiber), T2 (passive fiber), and R (reflected) for a coupler fabricated with two fibers twisted 5 times without holes.

It is crucial to note that the coupler was homemade, which may affect the reproducibility of the results. Nevertheless, when compared to conventional optical fibers, the affordability and simplicity of POF technology make it superior to conventional fibers in domestic networks. It is anticipated that, in the near future, POFs will experience a rise in popularity driven by the extensive deployment of fiber to the home (FTTH) and potentially within the home itself [3].

Slight variations during manufacturing may impact the signals at different output ports, particularly in terms of the coupling ratio. However, here, deliberate changes were introduced during manufacturing—specifically, variations in the torsion of the coupler—to assess the spectral influence of these changes at each output port. Figure 8 presents a spectral comparison of two couplers, with each spectrum measured at the T1, T2, and R output ports, manufactured by twisting the fiber ends two and five times, respectively.

The signal at each output port for both couplers is displayed in Figure 8a–c. Notably, the spectral traces measured at the respective ports are very similar for both couplers. These traces reveal that the number of twists does not significantly modify the spectral transmission at the output ports.

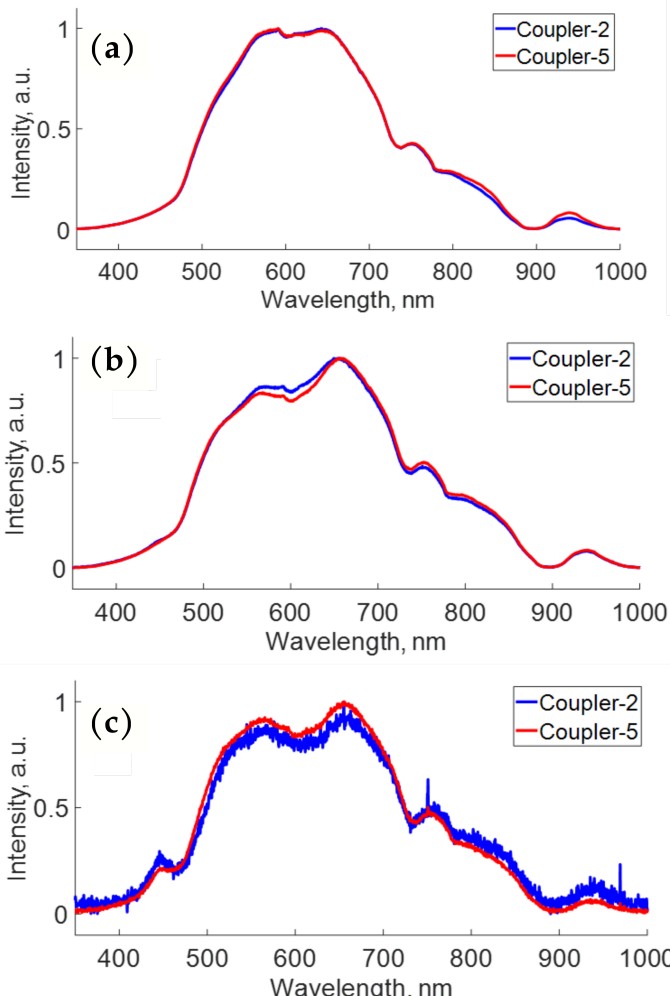

**Figure 8.** Spectral response for a coupler fabricated with two fibers without holes, twisted 2 and 5 times: (**a**) T1 (active fiber), (**b**) T2 (passive fiber), and (**c**) R (reflected).

Figure 9 illustrates the transmission measurements of the fiber optical filter (with two holes) for different values of the refractive index of a glucose solution filling the holes, comparing the results for both the straight-line configuration (Figure 9a) and in-loop mirror (POFLM) configuration (Figure 9b). In the straight-line configuration (Figure 9a), the transmission of the optical fiber with in-line holes increases with the rise in the refractive index, in line with previous findings [32]. Similarly, in the in-loop mirror (POFLM) configuration (Figure 9b), the transmission of the optical fiber with looped holes increases with an increase in the refractive index. Notably, in the latter case, the intensity for low concentrations (represented by the red and blue lines) shows significant variations, suggesting the possibility to discern low concentration values differing by only 1%, and the potential to lower the resolution and detection limit of the refractive index sensor system with the POFLM configuration.

A significant outcome when using the fiber optical filter is that the shape of the transmission spectrum remains unchanged despite variations in the refractive index in the holes. In other words, the refractive index variations only modify the intensity of the spectral waveform, and the bandpass filter remains unaltered. This lack of spectral

dependence on the refractive index contributes to the robustness, reproducibility, and feasibility of our proposed system.

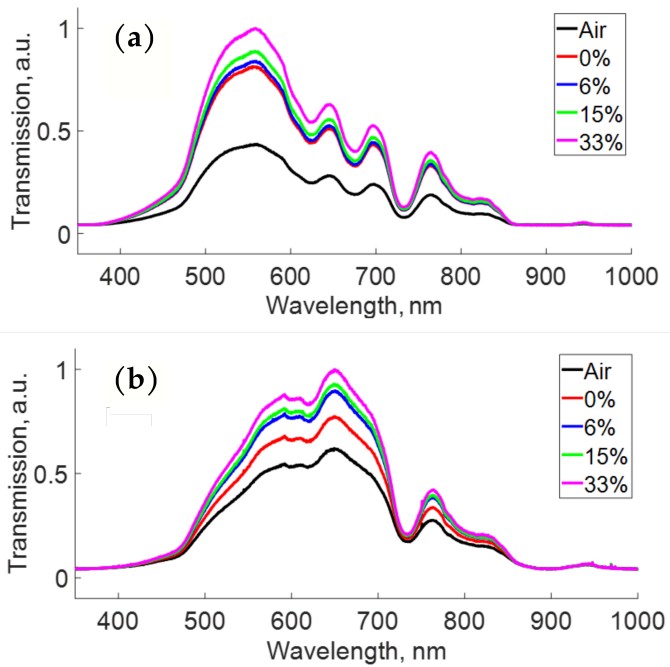

**Figure 9.** Transmission: (**a**) straight-line fiber and (**b**) POFLM fiber with two holes, for different glucose concentrations.

## 4. Conclusions

In this study, we have demonstrated the effectiveness of a plastic optical fiber with micro-holes to filter spectral signals in the visible region of the electromagnetic spectrum. The incorporation of holes in the POF enables the fabrication of bandpass spectral filters that are tunable by increasing the number of holes. This provides several key advantages, including ease of handling, high flexibility, robustness, straightforward fabrication and operation, cost-effectiveness, softness, and a light weight. Moreover, the POFLM configuration further enhances the tuning possibilities of the bandpass filter, although the reproducibility depends on the technique of fabrication. Whereas previous research has already highlighted the improved sensitivity of refractive index sensing with multiple holes in POFs, our work emphasizes that the fiber loop mirror configuration further enhances the sensitivity of such sensors using perforated POFs.

The intrinsic attenuation loss of common optical polymers is presented as a function of the wavelength [3]. Spectral variations can be achieved by altering the composition of the material used in the manufacturing of POFs. However, extrinsic losses can be introduced into the POF when altering its transmission through perforations, as proposed here for optical filtering. Since dispersion refers to the broadening of signals caused by the wavelength-dependent speed of propagation within the optical fiber, and it imposes constraints on the transmission over extended distances, we attribute the spectral response of our system to the modification of the modes by the holes, rather than to dispersion.

Furthermore, our results indicate that the plastic optical fiber is an excellent optical device for the development of robust, compact, and low-cost spectral filter technology. The compactness and robustness of our filter make it particularly promising for signal processing or sensing applications. However, we anticipate their applicability to other spectral regions, such as the C band, enabling us to leverage our device in fields like optical communications. Finally, we plan to embark on a theoretical study in the near future, wherein variables such as the wavelength, plastic fiber thickness, hole diameter and

number, refractive index, and temperature within the holes will be considered to provide a precise explanation of the phenomena present in our spectral filter.

**Author Contributions:** Conceptualization, H.S.-H. and A.M.-N.; methodology, H.S.-H. and A.B.-G.; software, J.F.-P. and B.B.-M.; writing—review and editing, H.S.-H., O.P. and B.B.-M.; formal analysis, A.M.-N., J.L.d.l.C.-G. and O.P. All authors have read and agreed to the published version of the manuscript.

**Funding:** This research was funded by Consejo Nacional de Humanidades, Ciencias y Tecnologías (CONAHCYT), grant number 319419.

**Institutional Review Board Statement:** Not applicable.

**Informed Consent Statement:** Not applicable.

**Data Availability Statement:** Data underlying the results presented in this paper are not publicly available at this time but may be obtained from the authors upon reasonable request.

**Acknowledgments:** The authors thank Guadalajara University, the Optical Research Center (CIO), and CONACYT.

**Conflicts of Interest:** The authors declare no conflicts of interest.

## Abbreviations

The following abbreviations are used in this manuscript:

| | |
|---|---|
| POF | Plastic Optical Fiber |
| PMMA | Polymethyl Methacrylate |
| WDM | Wavelength-Division Multiplexing |
| FBG | Fiber Bragg Grating |
| LPG | Long-Period Grating |
| SMF | Single-Mode Fiber |
| FOLM | Fiber Loop Mirror |
| POFLM | Plastic Optical Fiber Loop Mirror |
| CNC | Computer Numerical Control |

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
