# Peer review of "Plastic Optical Fiber Spectral Filter Based on In-Line Holes"

_photonics, doi:10.3390/photonics11040306_

Round 1

Reviewer 1 Report

Comments and Suggestions for Authors

The manuscript entitled “Plastic Optical Fiber Spectral Filter Based on in-Line Holes." by Azael Mora-Nunez et al. reports an investigation of highly reproducible spectral filters based on Plastic Optical Fiber with micro-holes. The author continues his study of this spectral filter by exploring fiber extended in a straight line and fiber optic loop mirror scheme configuration. Though Polymer or plastic optical fibers (POF) are potential topics that need investigation, the authors have not performed reasonable research on this investigation. Still, certain essential aspects, such as motivation for the work, novelty, theoretical background, losses, and potential application of these spectral filters, need to be discussed in this manuscript. Fiber loop mirror (FLM) has a well-established theory by Mortimore [1] and Morishita [2] in the 1550nm regime, so when incorporating FLM into any spectral filter of the visible range, the theoretical aspect of the spectral filter needs to be discussed.

1.    Mortimore, David B. "Fiber loop reflectors." Journal of lightwave technology 6, no. 7 (1988): 1217-1224.

2.    Morishita, K., and K. Shimamoto. "Wavelength-selective fiber loop mirrors and their wavelength tunability by twisting." Journal of lightwave technology 13, no. 11 (1995): 2276-2281.

 Also, the MZI-based spectral filter and the tuning with FLM have already been reported. [3]. Any spectral filter that can be tuned with FLM has been written by Naveen et al. [4].

 3.     Kumar, Naveen, and K. Ramachandran. "Mach–Zehnder interferometer concatenated fiber loop mirror-based gain equalization filter for an EDFA." Optics Communications 289 (2013): 92-96.

4.    Kumar, Naveen, and K. Ramachandran. "Dynamic spectral maneuvering by fiber Sagnac loop filter." Optics & Laser Technology 63 (2014): 144-147.

 The article is presented as documentation and needs some clarification. The language used in this article is reasonable. The critical statement to support the results/findings and the statement's citation should also be included. Overall, the article is well-intentioned but needs improvement. All the results are required to connect, and a thorough description by providing detailed information can improve the fineness of this manuscript. I want to address a few queries on this manuscript, which will help improve the quality of the article. Please find the comment below.

 1.    Page no 1, line 25, “Fiber optical filters play a crucial role in communication systems, particularly in wavelength-division multiplexing (WDM)”[**]. Fiber optical filters are crucial in communication systems [**], particularly in wavelength-division multiplexing (WDM). They are also widely employed in spectroscopy [**] and fiber optic sensing [**].Please provide suitable recent references if you have any.

2.    What is the strong motivation of this research work? It should include in the manuscript.

3.    Page no 1, line 29, FBG filters function as band rejection filters [**], whereas

LPG filters have the flexibility to be reconfigured into a bandpass filter [**]. Please provide suitable recent references if you have any.

4.    Page no 2, line 39, However, these devices are designed for use as sensors.[**]. Please provide suitable recent references if you have any.

5.    Page no 3, line 82, POF exhibits inherent losses attributed to factors such as absorption in the constituent material and Rayleigh scattering [**].Please provide suitable recent references if you have any.

6.    Page no 3, line 91, The first region spans approximately 470 nm to 600 nm, where the spectral transmission experiences a blue shift with an increasing number of holes in the POF. Why does the spectral transmission experience a blue shift?

  1. In Figures 3 and 4, Did the author check the stability of spectral response with 0 holes? Why is there a notch of ~620nm? 
  2. In Figure 3, ~770 nm, the transmission is higher with 3 holes than the 0 hole; why? 
  3. In Figure 4, ~770 nm, the transmission is higher with one hole than the 0 hole; why? 
  4. It is hard to compare the spectral response of both configurations in Figure 5. Did the author simulate the reactions, or is it experimental?  
  5. How does the author measure the spectral response shown in Figure 7?
  6. The spectral responses for air are shown in Figure 9, and they differ from the spectral response shown with 0 holes in Figures 3, 4, and 5. why

Author Response

The manuscript entitled “Plastic Optical Fiber Spectral Filter Based on in-Line Holes." by Azael Mora-Nunez et al. reports an investigation of highly reproducible spectral filters based on Plastic Optical Fiber with micro-holes. The author continues his study of this spectral filter by exploring fiber extended in a straight line and fiber optic loop mirror scheme configuration. Though Polymer or plastic optical fibers (POF) are potential topics that need investigation, the authors have not performed reasonable research on this investigation. Still, certain essential aspects, such as motivation for the work, novelty, theoretical background, losses, and potential application of these spectral filters, need to be discussed in this manuscript. Fiber loop mirror (FLM) has a well-established theory by Mortimore [1] and Morishita [2] in the 1550nm regime, so when incorporating FLM into any spectral filter of the visible range, the theoretical aspect of the spectral filter needs to be discussed.

  1.   Mortimore, David B. "Fiber loop reflectors." Journal of lightwave technology 6, no. 7 (1988): 1217-1224.
  2.   Morishita, K., and K. Shimamoto. "Wavelength-selective fiber loop mirrors and their wavelength tunability by twisting." Journal of lightwave technology 13, no. 11 (1995): 2276-2281.

Also, the MZI-based spectral filter and the tuning with FLM have already been reported. [3]. Any spectral filter that can be tuned with FLM has been written by Naveen et al. [4].

  1.     Kumar, Naveen, and K. Ramachandran. "Mach–Zehnder interferometer concatenated fiber loop mirror-based gain equalization filter for an EDFA." Optics Communications 289 (2013): 92-96.
  2.   Kumar, Naveen, and K. Ramachandran. "Dynamic spectral maneuvering by fiber Sagnac loop filter." Optics & Laser Technology 63 (2014): 144-147.

R: We appreciate your comments because they helped us to improve clarity in our proposed work. To help clarify the principle of operation, we added the references suggested (ref. 46-49), however, the reference [45] explains the influence of multiple modes in a gyroscope system. And a description is added in the results and discussion section (just before of Figure 5) to clarify the physical principle of our proposed system as follows:

“The transmittance of fiber loop reflectors based on single-mode fiber depends on the wavelength, coupling ratio, and loop birefringence  [46-49]. However, in our setup, we use a multimode fiber that considerably modifies and complicates such dependence. Reference [45] theoretically shows that the presence of multiple modes and mode coupling in the fiber loop degrades the fringe visibility and produces a residual phase error which affects the sensitivity and bias stability. In our experimental study, we take advantage of the fact that variations of refractive index in the loop of the Sagnac produce varying losses in the fiber core.”

The article is presented as documentation and needs some clarification. The language used in this article is reasonable. The critical statement to support the results/findings and the statement's citation should also be included. Overall, the article is well-intentioned but needs improvement. All the results are required to connect, and a thorough description by providing detailed information can improve the fineness of this manuscript. I want to address a few queries on this manuscript, which will help improve the quality of the article. Please find the comment below.

R: We appreciate your comments because they helped us to improve the fineness of our manuscript.

  1.   Page no 1, line 25, “Fiber optical filters play a crucial role in communication systems, particularly in wavelength-division multiplexing (WDM)”[**]. Fiber optical filters are crucial in communication systems [**], particularly in wavelength-division multiplexing (WDM). They are also widely employed in spectroscopy [**] and fiber optic sensing [**].Please provide suitable recent references if you have any.

R: Thank you for your comments, we really appreciate your suggestions. We include the  references [17], [18], [19] and [20], that enrich the state of the art of our work.

  1.   What is the strong motivation of this research work? It should include in the manuscript.

R: Thank you for your comments, we added one sentence about the motivation in the revised manuscript (end of introduction section) as follows:

The production of optical components based on fiber optic technology involves costly and labor-intensive processes, while silicon-based fiber technology tends to become delicate post-processing. Consequently, plastic optical fiber presents an incentive for us to devise cost-effective optical components that are practical for domestic use. Although its application is confined to short distances, it proves highly suitable for smaller-scale systems like automotive [38], textile sensors [39], medical [40] or home networking [41].

  1.   Page no 1, line 29, FBG filters function as band rejection filters [**], whereas

R:  Thank you for your comments. We include the reference [21] where the basic techniques for fiber grating fabrication, their characteristics, and the fundamental properties of fiber gratings are described. 

LPG filters have the flexibility to be reconfigured into a bandpass filter [**]. Please provide suitable recent references if you have any.

R:  Thank you for your comments. We include the reference [22] where a bandpass filter is demonstrated by fabricating an asymmetric long-period fiber grating (LPFG) in an off-set splicing fiber structure of two single mode fibers.

  1.   Page no 2, line 39, However, these devices are designed for use as sensors.[**]. Please provide suitable recent references if you have any.

R:  The statement in line 39, which refers to recent references [28], [29], and [30] in the revised version, has been omitted to prevent confusion.

  1.   Page no 3, line 82, POF exhibits inherent losses attributed to factors such as absorption in the constituent material and Rayleigh scattering [**].Please provide suitable recent references if you have any.

R:  Thank you for your comments. We include the reference [42] where the mechanical and optical properties of POFs relevant  to several applications are summarized.

  1.   Page no 3, line 91, The first region spans approximately 470 nm to 600 nm, where the spectral transmission experiences a blue shift with an increasing number of holes in the POF. Why does the spectral transmission experience a blue shift?

R:  Thank you for your comments. It is important to mention that we report experimental results. We believe that is necessary another theoretical study that includes the thickness of POF, the diameter and number of holes, refractive index in the holes, among others. Generally, we acknowledge that the phenomenon of total internal reflection is central to our system, as it is in all optical fibers, as mentioned in the paragraph that precedes figure 5.  However, in this regard, we include the following lines at the end of the conclusions:

However, we anticipate their applicability to other spectral regions, such as the C band, enabling us to leverage our device in fields like optical communications. Finally, we plan to embark on a theoretical study in the near future, wherein variables such as wavelength, plastic fiber thickness, hole diameter and number, refractive index, and temperature within the holes will be considered to provide a precise explanation of the phenomena present in our spectral filter.

  1. In Figures 3 and 4, Did the author check the stability of spectral response with 0 holes? Why is there a notch of ~620nm? 

R:  Thank you for your comments. Upon reviewing figures 3 and 4, we observe a notch at 620nm in the black curves (representing fibers without perforations). This phenomenon is attributed to attenuation losses, which are characteristic of PMMA-based plastic fibers, as reported in [3]. Notably, this valley persists in graphs depicting fibers with multiple perforations, thus ensuring the reproducibility and stability of our proposed system.

  1. In Figure 3, ~770 nm, the transmission is higher with 3 holes than the 0 hole; why? 

R:  Thank you for your comments. It is indeed a valid observation. Naturally, we anticipate lower transmission with three holes compared to no holes. However, it's important to note that the results were normalized with respect to their maximum to specifically compare only the spectral influence, as mentioned in the last sentence of the paragraph that precedes figure 3 as:

“It's essential to note that the traces are normalized, for the sake of comparison as fibers with more holes exhibit lower transmission intensity. ”

  1. In Figure 4, ~770 nm, the transmission is higher with one hole than the 0 hole; why? 

R:  Thank you for your comments. It's important to note that the results were normalized to specifically compare only the spectral influence. Moreover, the region around ~770 nm presents a complex behavior that we attribute to the interaction of multiple modes.  In this regard, we add a sentence at the end of the paragraph that precedes figure 4:

“Moreover, the region around ~770 nm presents a complex behavior that we attribute to the interaction of multiple modes.”

  1. It is hard to compare the spectral response of both configurations in Figure 5. Did the author simulate the reactions, or is it experimental?  

R:   It is important to mention that we report experimental results. We believe that it is necessary to perform in the future another theoretical study that includes the thickness of POF, the diameter and number of holes, refractive index in the holes, among others.

  1. How does the author measure the spectral response shown in Figure 7?

R:  Thank you for your comments. The paragraph after figure 6 describes how the ports were designated to perform the measurements as follows:

“For characterization purposes, we measured the spectrum at the output ports of a specially manufactured directional 2X2 coupler. The coupler was developed using the twisting technique of POF along approximately 5 cm, as illustrated in [16]. It's essential to note that one port of the coupler serves as the input for light. The output ports are designated as follows: the output port of the active fiber (the same input fiber) is marked as T1, the output port of the passive fiber (which receives the signal from the active fiber) is marked as T2, and the output port where the signal is reflected by Fresnel reflection is marked as R. The spectra measured at each port are depicted in Fig. 7.”

  1. The spectral responses for air are shown in Figure 9, and they differ from the spectral response shown with 0 holes in Figures 3, 4, and 5. why

R:  Thank you for your comments.  We add a sentence in each figure caption to specify the perforated fiber. Moreover, to clarify potential confusion in the graphs:

  1. a) The black traces in figures 3 and 4 do indeed represent the transmission of the fiber without perforations.
  2. b) Figure 5 does not depict the transmissions of the unperforated fiber. Instead, it illustrates the transmission of the fiber with 1, 2 or 3 perforations for both the straight-line and POFLM configurations, respectively.
  3. c) Figure 9 illustrates the scenario where the fiber possesses 2 perforations: a) in a straight line, and b) in the POFLM configuration, respectively.

Reviewer 2 Report

Comments and Suggestions for Authors

line 1: Plastic Optical Fiber > plastic optical fiber (POF)

line 15-21: the introduction should shortly mention and segregate some of following concepts as they form boundary conditions of the concept of individual in-line holes

- optical fibers (or waveguides) with stochastic or random implementation of scatterers or parametrized multi-type disorders

- optical fibers (or waveguides) with 'inverse holes' (i.e. 'empty' voids and colloidal dielectric) 

line 58: It would be beneficial for the interdisciplinary reader if the concept of "POF with n in-line holes" would be clearly visually introduced e.g. in a panel of Figure 1. Possibly it is enough to write an additional label to perforated fiber if these terms are technically equivalent. Please enlarge the zoom of the perforated bars a bit.

line 82-86: A clear statement how in particular the given fluctuations in density, composition and orientation qualitatively and possibly quantitatively affect absorption, transmission and scattering would be interesting for the interdisciplinary reader.

line 96-97: ", a second peak displays a decrease in transmission intensity of 0.963, 0.792, 0.597, and 0.438 for 0, 1, 2, and 3 holes, respectively." > I detect a bit of unnecessary ambiguity in this sentence. Possibly, consider a small table 0, 1, 2, 3 holes and the respective T values. Please check the usage of the prepositions "in", "of" here.

Figure 3 & 4: Please correct typo (unnecessary blank) in legend

line 185: unknown terminus technicus: 'shaoe'

line 202-205: The conclusion deserves a bit more expansion, e.g. the comparison of the performance of POF with in-line holes to other fiber concepts as e.g. mentioned in the expansion of the introduction.

#

Author Response

line 1: Plastic Optical Fiber > plastic optical fiber (POF)

R: Thank you for your comments, we make the suggested changes.

line 15-21: the introduction should shortly mention and segregate some of following concepts as they form boundary conditions of the concept of individual in-line holes

- optical fibers (or waveguides) with stochastic or random implementation of scatterers or parametrized multi-type disorders

- optical fibers (or waveguides) with 'inverse holes' (i.e. 'empty' voids and colloidal dielectric) 

R: We appreciate your comments because they helped us to improve clarity in our proposed work. We agree in the introduction section that discussing disorders present in optical fibers in lines 51-62 of the revised version as follows:

Some theoretical studies have demonstrated that morphological disorders in waveguides can significantly impact the transmitted optical signal. For instance, simulations in [34] examined the transmission spectra of photonic components with up to four types of morphological disorder as a materials parameter. The results showed that various types of disorder, as well as combinations of two types of disorder, can be intentionally utilized to adapt the spectral transmission characteristics. Consequently, these findings may inform the design of small-footprint photonic components such as scatterers, diffusers, and waveguides. Additionally, in [35], researchers demonstrated that light transport in a disordered ensemble of resonant atoms placed in a waveguide is highly sensitive to the cross-sectional sizes of the waveguide. Specifically, the transmittance magnitude was found to undergo complex, non-monotonic dependence on the transverse sizes of the waveguide, highlighting the intricate nature of light propagation in disordered systems.

Furthermore, in the Results and Discussion section, we have added a description regarding attenuation and dispersion in lines 111-118 of the revised version as follows:

“The transmission quality of an optical fiber is defined by two properties: attenuation and dispersion. The sources of attenuation in optical fibers can indeed be categorized into two types: intrinsic and extrinsic. Intrinsic sources are inherent to the material, such as material absorption and Rayleigh scattering, while extrinsic sources are introduced during the manufacturing of the fiber, including structural imperfections and microbends. Dispersion refers to the broadening of signals caused by the wavelength-dependent speed of propagation within the optical fiber, and it imposes constraints on transmission over extended distances.

POF exhibits inherent losses attributed to factors such as absorption in the constituent material and Rayleigh scattering [42]. These losses arise from molecular vibrational absorption of groups like C-H, N-H, and O-H, absorption due to electronic transitions within molecular bonds, and scattering resulting from fluctuations in composition, orientation, and density [43, 44]. However, extrinsic losses can be introduced into POF by altering its transmission through perforations, as proposed by the authors for optical filtering.”

line 58: It would be beneficial for the interdisciplinary reader if the concept of "POF with n in-line holes" would be clearly visually introduced e.g. in a panel of Figure 1. Possibly it is enough to write an additional label to perforated fiber if these terms are technically equivalent. Please enlarge the zoom of the perforated bars a bit.

R: We are grateful for your comments and value your suggestions. In response, we have added supplementary labels to the panel of Figure 1 and revised certain sentences to enhance the clarity of the concept.

line 82-86: A clear statement how in particular the given fluctuations in density, composition and orientation qualitatively and possibly quantitatively affect absorption, transmission and scattering would be interesting for the interdisciplinary reader.

R: We are grateful for your comments and value your suggestions. That's correct. The transmission quality of optical fibers is indeed determined by attenuation and dispersion. In this regard, we include the following lines in the revised version, in the first paragraph of Results and Discussions section:

“The transmission quality of an optical fiber is defined by two properties: attenuation and dispersion. The sources of attenuation in optical fibers can indeed be categorized into two types: intrinsic and extrinsic. Intrinsic sources are inherent to the material, such as material absorption and Rayleigh scattering, while extrinsic sources are introduced during the manufacturing of the fiber, including structural imperfections and microbends. 

POF exhibits inherent losses attributed to factors such as absorption in the constituent material and Rayleigh scattering [42]. These losses arise from molecular vibrational absorption of groups like C-H, N-H, and O-H, absorption due to electronic transitions within molecular bonds, and scattering resulting from fluctuations in composition, orientation, and density [43, 44]. However, extrinsic losses can be introduced into POF by altering its transmission through perforations, as proposed by the authors for refractive index sensing.”

line 96-97: ", a second peak displays a decrease in transmission intensity of 0.963, 0.792, 0.597, and 0.438 for 0, 1, 2, and 3 holes, respectively." > I detect a bit of unnecessary ambiguity in this sentence. Possibly, consider a small table 0, 1, 2, 3 holes and the respective T values. Please check the usage of the prepositions "in", "of" here.

R: Thank you for your comments, we really appreciate your suggestions. We make the suggested changes.

Figure 3 & 4: Please correct typo (unnecessary blank) in legend

R: Thank you for your comments, we make the suggested changes.

line 185: unknown terminus technicus: 'shaoe'

R: Thank you for your comments, we replace “shaoe” by shape. We check the entire article to find spelling errors.

line 202-205 The conclusion deserves a bit more expansion, e.g. the comparison of the performance of POF with in-line holes to other fiber concepts as e.g. mentioned in the expansion of the introduction.

R:  Thank you for your comments, we agree that the conclusions section would benefit from additional expansion. We added a paragraph in the conclusions section, on lines 237-244 of the revised version:

“The intrinsic attenuation loss of common optical polymers is presented as a function of wavelength [3]. Spectral variations can be achieved by altering the composition of the material used in the manufacturing of POFs. However, extrinsic losses can be introduced into POF by altering its transmission through perforations, as proposed by the authors for optical filtering. Since dispersion refers to the broadening of signals caused by the wavelength-dependent speed of propagation within the optical fiber, and it imposes constraints on transmission over extended distances, we attribute the spectral response of our system to the modification of the modes by the holes, rather than to dispersion.”

Reviewer 3 Report

Comments and Suggestions for Authors

The paper presents a study on a spectral filter based on plastic optical fiber with in-line micro-holes. The authors explore the fabrication of such filters using a Computer Numerical Control (CNC) mounted drill for 3D printing. Two configurations of the filter are investigated: a straight-line fiber extension and a fiber optic loop mirror scheme. The transmission characteristics of the filter with varying numbers of holes and distances between holes are evaluated. However, this paper discusses the fabrication and testing of filters with in-line holes, the overall objective or significance of this work in the context of existing literature remains unclear. The use of a CNC-mounted drill to create micro-holes in the plastic optical fiber is not particularly innovative or unique. Moreover, while this paper presents experimental results on the transmission characteristics of the filters with varying parameters, the depth of analysis seems insufficient. Therefore, I do not strongly recommend the publication of this paper. Besides, some aspects of this paper require improvement:

1.      Figures 6 to 9 do not seem to specify which type of fiber with holes is being used.

2.      The English text of this paper contains some typographical errors that need to be proofread. For example, in the conclusion, "turther" should be corrected to "further."

Comments on the Quality of English Language

The English text of this paper contains some typographical errors that need to be proofread. For example, in the conclusion, "turther" should be corrected to "further."

Author Response

The paper presents a study on a spectral filter based on plastic optical fiber with in-line micro-holes. The authors explore the fabrication of such filters using a Computer Numerical Control (CNC) mounted drill for 3D printing. Two configurations of the filter are investigated: a straight-line fiber extension and a fiber optic loop mirror scheme. The transmission characteristics of the filter with varying numbers of holes and distances between holes are evaluated. However, this paper discusses the fabrication and testing of filters with in-line holes, the overall objective or significance of this work in the context of existing literature remains unclear. The use of a CNC-mounted drill to create micro-holes in the plastic optical fiber is not particularly innovative or unique. Moreover, while this paper presents experimental results on the transmission characteristics of the filters with varying parameters, the depth of analysis seems insufficient. Therefore, I do not strongly recommend the publication of this paper. Besides, some aspects of this paper require improvement:

  1.     Figures 6 to 9 do not seem to specify which type of fiber with holes is being used.

R: Thank you for your comments, we add a sentence in each figure caption to specify the fiber type.

  1. The English text of this paper contains some typographical errors that need to be proofread. For example, in the conclusion, "turther" should be corrected to "further."

R: Thank you for your comments, We check the entire article to find spelling errors.

Reviewer 4 Report

Comments and Suggestions for Authors

The proposed article concerns a POF-based filter in the visible spectrum as a low-cost, robust and reproducible device. It is well written and provides a lot of experimental measurements with consideration. However, some issue in the article structure and some lack of information led me to suggest a major revision to improve the manuscript:

1. there are too much references in the introduction. In half a page of text, almost 30 references were used. In my opinion is too much: consider either to reduce the number of referenes, or to enlarge the text motivating this choice.

2. more background on holes in POF (as well as in-line holes in POF). Since it is the most important technology to be introduced for the manuscript understanding, I suggest to spend more sentences on this topic. 

3. it is not clear to me the importance to use The Plastic Optical Fiber Loop Mirror (POFLM) in the experimental setup. Please explain better the characteristics of this device and the advantage on using it for the measurements you propose.

4. the second paragraph (2. Experimental Setup) is too short. Please provide more information on what are you doing, how and why.

5. Is there any temperature shifting due to the thermo-optical effect for the filter spectrum? was it considered?

6. is it possible to use the same technology to design a filter in C band?

7. what do you mean for straight-line scheme configuration? Consider to use some block scheme for a better understanding.

8. a coupler homemade whas mentioned, without explaining its characteristics and how it was made. Please provide more information.

9. Is it also possible to use this structure as glucose sensor?

Author Response

The proposed article concerns a POF-based filter in the visible spectrum as a low-cost, robust and reproducible device. It is well written and provides a lot of experimental measurements with consideration. However, some issue in the article structure and some lack of information led me to suggest a major revision to improve the manuscript:

  1. there are too much references in the introduction. In half a page of text, almost 30 references were used. In my opinion is too much: consider either to reduce the number of referenes, or to enlarge the text motivating this choice.

R: We appreciate your comments because they helped us to improve clarity in our proposed work. We opted to condense the references, though additional ones were included at the request of other reviewers.

  1. more background on holes in POF (as well as in-line holes in POF). Since it is the most important technology to be introduced for the manuscript understanding, I suggest to spend more sentences on this topic. 

R: We appreciate your comments. Several studies have explored processed plastic fibers; however, there is a notable scarcity of research on perforated plastic fibers. In this regard, we have added a brief discussion in the introduction section at lines 37-40 of the revised version:

Generally, methods for POF structural modification include tapering [28], side-polishing [29], screw-shaped [30], spiral-shaped [31] and multi-notched [32]. Moreover, there were very few studies with holes in fiber [33,36,37]. Specially, techniques for creating holes in POFs involve the use of a drill bit [33,36], and a mechanical die press print method [37].

  1. it is not clear to me the importance to use The Plastic Optical Fiber Loop Mirror (POFLM) in the experimental setup. Please explain better the characteristics of this device and the advantage on using it for the measurements you propose.

R: We appreciate your comments because they helped us to improve clarity in our proposed work. The phrasing might not sufficiently emphasize the significance of the POFLM setup. Consequently, we have reorganized the description within the experimental arrangement section. Furthermore, we add some sentences in the revised version:

Lines 154-157: Moreover, leveraging the additional degrees of freedom in the mirror configuration compared to the straight-line setup, and taking advantage of the experience acquired in previous work [16], we analyzed the response with perforations in the loop of POFLM as shown in Fig. 5, see Fig. 1(b).

Lines 248-253: Moreover, the POFLM configuration further enhances the tuning possibilities of the bandpass filter, although the reproducibility depends on the technique of fabrication. Whereas previous research has already highlighted the improved sensitivity of refractive index sensing with multiple holes in POF, our work emphasizes that the fiber loop mirror configuration further enhances the sensitivity of such sensors using perforated POF.

  1. the second paragraph (2. Experimental Setup) is too short. Please provide more information on what are you doing, how and why.

R: We appreciate your comments because they helped us to improve clarity in our proposed work. In this regard, we have inserted additional text into several lines of the revised version:

Line 70-71: Both configurations are introduced to showcase the capabilities of the proposed spectrum filter utilizing perforated plastic fibers.

Line 86-90: The interrogation system consists of a tungsten lamp that serves as a broadband light source, and an Ocean Optics spectrometer (model USB2000CXR) that aids us in measuring the optical signals at the output of our filters. Thus, the characterization of the spectral filter is conducted by measuring transmittance as a function of the number of holes.

91-104:Additionally, the sensing of refractive index is examined by filling the holes with dextrose (D-Glucose, J. T. Baker) concentrations, ranging from distilled water to saturation. The specific dextrose solutions cover a density range from 0 g/100ml (pure distilled water) to 33 g/100ml, with intermediate concentrations at intervals of 3 g/100ml. Moreover, introducing variations in the refractive index within the holes enables us to discern the filter's response under diverse conditions. In essence, modifications in the frequency or intensity of the signal by variations of refractive index will afford us enhanced utilization of the proposed spectral filter. The solutions, ranging from the lowest to the highest refractive index, were introduced into the holes. Each solution was carefully poured into a small container containing the holes. Due to the challenging removal of the highest index solution from the hole, extra care was taken for accurate results. After measurement, the liquid was extracted using a syringe, absorbed with paper, and then dried by blowing air, ensuring the output power matched that of the hole filled with air. This process was repeated for each solution.

  1. Is there any temperature shifting due to the thermo-optical effect for the filter spectrum? was it considered?

R: We appreciate your comments, The thermo-optic effect was not assessed in this study; all measurements were conducted at room temperature. Nevertheless, we are compelled to undertake a future theoretical investigation wherein we incorporate additional variables such as the thermo-optic effect, as suggested to us.

  1. is it possible to use the same technology to design a filter in C band?

R: We appreciate your comments because they helped us to improve clarity in our proposed work. The results presented in this study are concentrated solely on the visible region. However, we anticipate their applicability to other spectral regions, such as the C band, enabling us to leverage our device in fields like optical communications. We aim to conduct a theoretical investigation in the future, analyzing the impact of wavelength, thickness of POF, the diameter and number of holes, refractive index in the holes, among others on the proposed spectral filter. In this regard, we include the following lines at the end of the conclusions:

However, we anticipate their applicability to other spectral regions, such as the C band, enabling us to leverage our device in fields like optical communications. Finally, we plan to embark on a theoretical study in the near future, wherein variables such as wavelength, plastic fiber thickness, hole diameter and number, refractive index, and temperature within the holes will be considered to provide a precise explanation of the phenomena present in our spectral filter.

  1. what do you mean for straight-line scheme configuration? Consider to use some block scheme for a better understanding.

R= Thank you for your comments, we really appreciate your suggestions. We make the suggested changes. We added a better description on the label of figure 1.

“Schematic diagram Fiber optical filters: (a) straight-line configuration, (b) POFLM configuration. ”

  1. a coupler homemade whas mentioned, without explaining its characteristics and how it was made. Please provide more information.

R: Thank you for your comments. Certainly, the couplers were homemade, leveraging the expertise gained from a prior study [16]. In this regard, we mention, in lines 193-194, the reference where the manufacturing process is described with precision as follows:

“The coupler was developed using the twisting technique of POF along approximately 5 cm, as illustrated in [16].”

  1. Is it also possible to use this structure as glucose sensor?

R: Thank you for your question. Certainly, this structure has the potential to function as a glucose sensor. Shin, J. D., & Park, J., have previously suggested in [33] utilizing POF with in-line hole as a refractive index sensor. Moreover, In both the comments for Figure 9 and in the conclusion, we emphasized that the loop-mirror configuration has the potential to decrease the detection limit and attain higher resolution than previously documented.

Round 2

Reviewer 1 Report

Comments and Suggestions for Authors

The revised manuscript entitled “Plastic Optical Fiber Spectral Filter Based on in-Line Holes." by Azael Mora-Nunez et al. reports an investigation of highly reproducible spectral filters based on Plastic Optical Fiber with micro-holes. The authors have significantly improved the review manuscript by illustrating their research work regarding modifications, figures and contents. Also, the authors have given satisfactory responses to the comment raised. The article will be the appropriate form for publication in Photonics after addressing the following comment.

1.     On page no 5, line 166, “Moreover, the region around 770 nm presents a complex behavior that we attribute to the interaction of multiple: modes [**]. Please provide suitable reference if authors have any.

Author Response

On page no 5, line 166, “Moreover, the region around 770 nm presents a complex behavior that we attribute to the interaction of multiple: modes [**]. Please provide suitable reference if authors have any.

R:  Thank you for your comments. We include references [45,46] where the optical distribution and operational principles of various devices based on multimode interference are discussed.

Finally, we would like to express our sincere gratitude to the reviewer for dedicating their time to evaluate our manuscript. We hope that the revised manuscript version meets the requirements of the reviewer.

Reviewer 3 Report

Comments and Suggestions for Authors

The author has addressed all of my comments. I believe this paper is ready for publication.

Author Response

Dear reviewer, We value the time and effort you dedicated to reviewing our work and contributing to its improvement.

Reviewer 4 Report

Comments and Suggestions for Authors

The authors have provided exaustive responses. In my opinion the article is suitable for publication.

Author Response

(The authors gave the same response as above.)
